# Updated Erdman reveals tandem repeat copy number is phase-variable and impacts *M. tuberculosis* adaptation across evolutionary timescales

Samuel J. Modlin,[1] Nachiket Thosar,[1] Paulina M. Mejía-Ponce,[1] Raegan L. Lunceford,[1] Gaelle Guiewi Makafe,[2] Brian Weinrick,[2] Faramarz Valafar[1]

ABSTRACT   High-quality reference genomes are essential for comparative genomics and accurate genotype-phenotype mapping. Here, we corrected the *Mycobacterium tuberculosis* Erdman strain reference genome (Erdman_TI) using ultra-deep HiFi sequencing. Among the small variants (*n* = 275) between Erdman_TI and the current Erdman reference NC_020559.1 (Erdman_STJ), numerous are likely errors in Erdman_STJ. We identified a novel bias toward in-frame structural variations (SVs) in *pe/ppe* genes and 28 SVs between Erdman_TI and Erdman_STJ, half representing likely errors in Erdman_STJ. Other SVs were consistent with *in vitro* evolution, including copy number variation (CNV) of promoter tandem repeats (PTRs). PTR CNVs were polyphyletic and within isogenic populations ($10^{-2}$–$10^{-3}$ CNVs/chromosome), demonstrating the impact of phase-variable CNV across evolutionary timescales. These hypervariable PTRs pinpoint a genomic basis for rapidly switching nitric oxide resistance (Dop), biofilm formation (LpdA), drug tolerance (EfpA), and glycerol utilization (GlpD2) phenotypes. This work uncovers a common phase variation mechanism obscured by short-read sequencing limitations and provides an improved reference for comparative studies.

IMPORTANCE *Mycobacterium tuberculosis* (*Mtb*), the pathogen responsible for tuberculosis, is often described as genetically stable. Our findings reveal an overlooked evolutionary adaptation mechanism: phase variation driven by tandem repeat copy number changes in gene promoters. Enabled by ultra-deep, long-read sequencing, we corrected errors in the Erdman reference genome and uncovered frequent, spontaneous expansions and contractions of promoter repeats upstream of genes linked to nitric oxide resistance, drug efflux, and biofilm formation. Through altering promoter strength, these dynamic promoter variants may generate phenotypic diversity within subpopulations and across diverse clinical lineages, suggesting a conserved evolutionary advantage for navigating host-imposed stress. This reframes *Mtb*'s evolutionary potential, highlighting how adaptive flexibility has been underestimated due to reliance on short-read sequencing and limited resolution of subpopulations at standard genomic depths. Our findings underscore the need to integrate structural variation-aware approaches into studies of *Mtb* pathogenesis, evolution, and drug response.

KEYWORDS   *Mycobacterium tuberculosis*, phase variation, tandem repeat, copy number, stress adaptation, evolution

Reference genomes are an indispensable research tool, offering a foundational basis for genome comparison and analysis. While extant reference genome assemblies are largely accurate, advances in sequencing read length and accuracy over the past decade have highlighted the limitations of earlier methods (1–3). Even highly accurate reference

**Peer Reviewers** William R. Jacobs, Albert Einstein College of Medicine, Bronx, New York, USA; Conor Meehan, Nottingham Trent University, Nottingham, United Kingdom

Address correspondence to Faramarz Valafar, faramarz@sdsu.edu.

Samuel J. Modlin and Nachiket Thosar contributed equally to this article. Author order was determined alphabetically by last name.

The authors declare no conflict of interest.

See the funding table on p. 20.

genome assemblies omit essential genetic variation present in subpopulations of the strain of interest, a critical component of bacterial adaptive capacity. Subpopulations factor critically in a strain's capacity for phenotypic plasticity, significantly influencing adaptability and survival (4). Moxon and colleagues (5) introduced the concept of "mutable contingency loci" in pathogenic bacteria, proposing that hypervariable genetic elements accelerate adaptive evolution. These contingency loci are not happenstantial features that emerge sporadically; they are prone to mutate by their biophysical nature and thus represent a feature of the strain or species that harbor them. Reference genomes with such loci identified would provide a more complete picture of adaptive capacity and would be more faithful to biological reality—where heterogeneity prevails (6)—than the conventional static consensus genome. Deep sequencing with highly accurate long reads can accomplish this by confidently detecting structural variation (SV) in rare subpopulations that evade characterization by short-read sequencing methods (7). Here, we apply ultra-deep sequencing with HiFi reads—a highly accurate SV-aware sequencing technology (1)—to a primary reference strain of major human pathogen *Mycobacterium tuberculosis* (*Mtb*) to improve its reference genome assembly and capture its subpopulation composition, enabling a more comprehensive view of its genomic complexities and microevolutionary dynamics.

*Mycobacterium tuberculosis*, the causative agent of human tuberculosis (TB), is widely studied due to its significant impact on public health (8). As *Mtb* has co-evolved with humans, it has developed sophisticated mechanisms to modulate host immune responses, making it an exceptionally deadly pathogen. This long-standing evolutionary relationship has provoked *Mtb* to fine-tune its genetic and epigenetic traits to thrive within the human pulmonary environment. *Mtb* laboratory reference strains play a crucial role in studying TB pathophysiology and identifying potential targets for clinical interventions. One of the most commonly used *Mtb* reference strains, Erdman (ATCC 35801), was originally isolated from human sputum in 1945 at the Mayo Clinic in Rochester, Minnesota, USA (9). The Erdman strain has been investigated extensively in studies of virulence (10), immunization (11), antibiotic resistance (12), and comparative genomics (13). Such studies have been performed in multiple animal models, including non-human primates, guinea pigs, mice, and rabbits, with comparative results demonstrating that *Mtb* Erdman is highly virulent (10). Erdman-infected animal TB models display a significant disease burden, including higher bacterial loads in the spleen and lung, larger necrotic granulomas, increased systemic inflammation resulting in a spike in macrophage and T cell activity (10), and cases where prolonged infection advanced into chronic conditions of the lung and spleen (13). Genotypic investigation comparing Erdman and H37Rv has yielded notable discoveries, including heterogeneity between the strains in the RD6 region (13).

The current Erdman reference genome (NC_020559.1) was sequenced in 2012 through a combination of Roche (GS FLX Titanium sequencer), Illumina (Genome Analyzer IIx), and ABI 3730xl sequencing technologies and assembled (GS *De Novo* Assembler version 2.6) into scaffolds and contigs by researchers in Shinjuku, Tokyo, Japan (9). Sanger sequencing of PCR fragments was used to fill the gaps of the scaffold and complete the assembly. We refer to this assembly henceforth as "Erdman$_{STJ}$." While assemblies produced through these technologies are largely correct, sequencing technology has advanced dramatically in the past decade, particularly in read length and accuracy (1). Recent long-read reference genome assemblies have corrected errors in assemblies generated from first- and second-generation sequencing technologies (3). We previously corrected the reference genome of the *Mtb* avirulent type strain H37Ra (14), reducing the set of genes with variants compared with its virulent sister strain H37Rv by more than half. Like the H37 strains, Erdman is frequently used for forward genetic screens (15, 16), transcriptional adaptation (17), and experimental evolutionary studies (18) that require mapping short-read DNA or RNA sequencing data back to a reference genome. Errors in the reference sequence can propagate systematically into such downstream studies, potentially occluding important evolution or transcriptional

responses and resulting in variants that are attributed spuriously as adaptation to experimental conditions rather than being recognized as errors in the reference.

Bacterial populations often exhibit genetic and phenotypic diversity within clonal populations (19). This phenotypic flexibility, often mediated through phase-variable elements, is a pivotal attribute shared by many pathogenic bacteria, facilitating rapid adaptation to new challenges (20). First observed in flagellar antigen switching in *Salmonella* over a century ago (21), "phase variation" refers to the reversible, high-frequency switching of phenotypic traits and represents a widespread adaptive strategy among bacteria (22). Phase variation generates heterogeneous clonal populations in which individual cells stochastically switch phenotypes, facilitating survival through bet-hedging in fluctuating environments. Phase variation is especially common in pathogens and aids in evading host immunity (20), persisting through sudden introduction of host-imposed stress, and opportunistic pathogenicity (23). Multiple molecular mechanisms underlie phase variation, including site-specific DNA inversions where specialized recombinases flip DNA segments, slipped-strand mispairing at hypermutable simple sequence repeats causing frameshifts or affecting promoter spacing, gene conversion and recombination events that shuffle genetic sequences, and epigenetic mechanisms through DNA methylation and transcription factor interactions at promoters, or phase-variable DNA methyltransferases (22). Each mechanism achieves reversible ON/OFF or allele switching at frequencies much higher than typical mutation rates, providing a repertoire of phenotypic subpopulations without permanent genomic change. This contingency strategy is particularly valuable for traits under intermittent selection, such as surface antigens facing immune pressure or genes needed only in certain environmental niches or nutrient constraints (5).

*M. tuberculosis* has traditionally been viewed as genomically static, with a clonal population structure and relatively slow mutation rate compared to other bacterial pathogens (24). However, recent advances in sequencing technologies have revealed unexpected genomic plasticity in *Mtb*, uncovering several mechanisms of phase variation that generate phenotypic diversity. Growing evidence now shows that *Mtb* employs phase variation through homopolymeric tract slippage with important phenotypic consequences. These include *glpK* frameshifts that confer transient drug tolerance (25) and variations in the ESX-1 secretion system that modulate host-pathogen interactions (26, 27). Variable number tandem repeats (VNTRs) have been recognized in the *Mtb* genome for decades (28) but have been primarily exploited epidemiologically as markers for strain differentiation (29) rather than investigated as potential drivers of phenotypic variation.

In this work, we present a corrected reference genome for the Erdman strain, "Erdman$_{TI}$" ("TI" for Trudeau Institute) from the Trudeau Mycobacterium Culture Collection (TMC 107, deposited in ATCC as ATCC 35801). This strain was prepared from a low-passage stock and was not passaged after isolation before being grown for DNA extraction. The extracted DNA was ultra-deep-sequenced (mean coverage depth = 6,349) with highly accurate (1) HiFi reads to produce a finished-grade reference genome. This updated genome enabled us to reassess genomic differences between Erdman and other key *Mtb* strains, such as H37Rv, and uncovers novel structural and small variants with implications for *Mtb* pathogenicity and drug tolerance. A large yet frequently unacknowledged imitation of nearly all sequencing studies is that they sample a minuscule fraction of the bacterial population, sampling at depth sufficient only to capture subpopulations at frequencies of ~1%. Here, through ultra-deep sequencing, we demonstrate that tandem repeat copy number variations (CNVs) are a common form of phase variation in *Mtb* that creates subpopulations with distinct clinical phenotypes. These findings underscore the importance of leveraging SV-aware sequencing technology to update bacterial reference genomes and the utility of ultra-deep sequencing for identifying frequently emerging subpopulations.

## RESULTS AND DISCUSSION

### Reference update

#### *HiFi sequencing enables an improved Erdman reference genome assembly (Erdman$_{TI}$)*

Erdman strain from the Trudeau Mycobacterial Culture Collection (TMC 107, deposited in ATCC as ATCC 35801) was subcultured, and extracted DNA was sequenced on the PacBio Revio system with circular consensus sequencing to produce high-fidelity consensus sequencing reads known as "HiFi" reads (1). Ultra-deep sequencing of Erdman produced 2,796,649 HiFi reads with median read length = 10,515, $N_{50}$ = 11,138, and a median quality value of 32.4. HiFi reads were assembled, polished, and circularized (Methods and Materials) into a single contiguous sequence (4,414,920 bp) containing 4,160 genes with an average coverage depth of 6,349×. The updated Erdman reference genome assembly is available on NCBI (BioProject No. PRJNA1250540). The Erdman$_{STJ}$ and Erdman$_{TI}$ genome assemblies had an average sequence identity of 99.97%. Calling small variants between the consensus sequences identified 275 variants between Erdman$_{STJ}$ and Erdman$_{TI}$ (Table S1) across 73 genes (Fig. 1A), resulting in truncation or frameshift of 32 genes relative to Erdman$_{STJ}$.

Variants between the Erdman assemblies were over 18-fold more prevalent in *pe/ppe* family genes than other genes (odds ratio = 18.79, $P = 4.915 \times 10^{-22}$). *pe/ppe* genes are hypervariable among *Mtb* strains and contain highly repetitive sequences with a high degree of homology between genes (30). These features result in both a high rate of genuine variation and make these genes challenging to assemble using short reads, as they do not span large enough genomic regions to confidently map to repetitive and homologous regions of the genome (31).

Next, we evaluated the concordance of variants between the Erdman genomes to *Mtb* virulent type strain H37Rv. We called variants in an H37Rv long-read assembly against Erdman$_{TI}$ and Erdman$_{STJ}$, as well as between the two Erdman genomes. We then assessed the relative proportion of variants common to H37Rv and one of the two Erdman genomes but discordant with the other (Fig. 1B). Like Erdman, H37Rv belongs to Lineage 4, but is located in a different clade in the phylogenetic tree. Therefore, variants evolved *in vitro* between different Erdman subcultures and genuine variants should match each Erdman sequence in a similar proportion. In contrast, sequencing errors in one Erdman assembly would nearly always match H37Rv in the correct assembly (since H37Rv and Erdman share 99.97% sequence identity). While only eight variants were common to H37Rv and Erdman$_{STJ}$ with respect to Erdman$_{TI}$ (Table S2), 198 variants across 60 genes with respect to Erdman$_{STJ}$ were common to H37Rv and Erdman$_{TI}$ (Fig. 1B and Table 1; Table S1). Among variants between Erdman$_{TI}$ and Erdman$_{STJ}$, H37Rv was over 85 times more likely to match the genotype of Erdman$_{TI}$ than Erdman$_{STJ}$ (odds ratio = 85.8, $P = 1.6 \times 10^{-72}$), demonstrating that Erdman$_{TI}$ corrects numerous errors in Erdman$_{STJ}$. While a subset of variants in Erdman$_{STJ}$ with respect to both H37Rv and Erdman$_{TI}$ is likely genuine, the massive discrepancy in H37Rv-concordant variants implies that the majority are errors in the Erdman$_{STJ}$ assembly.

Variants shared by H37Rv and Erdman$_{TI}$ with respect to Erdman$_{STJ}$ largely comprised *pe_pgrs* genes (145/198, 73.2%). This suggests that long-read HiFi sequencing better resolved the highly repetitive and homologous regions present in *pe_pgrs* genes, consistent with previous reports (14, 31, 32). Five of the remaining variants were intergenic, and the remaining 48 variants fell within genes. These 48 intragenic variants comprise 32 distinct genes, including 8 with nonsynonymous single nucleotide polymorphisms (nsSNPs) and 14 with that induce frameshifts (Fig. 1C). These likely sequencing errors in Erdman$_{STJ}$ include several notable genes (Fig. 1C), such as virulence-regulating two-component system member *phoP* (33, 34) and a frameshift-inducing 1-bp insertion in efflux pump *jefA,* whose overexpression confers isoniazid and ethambutol resistance (35).

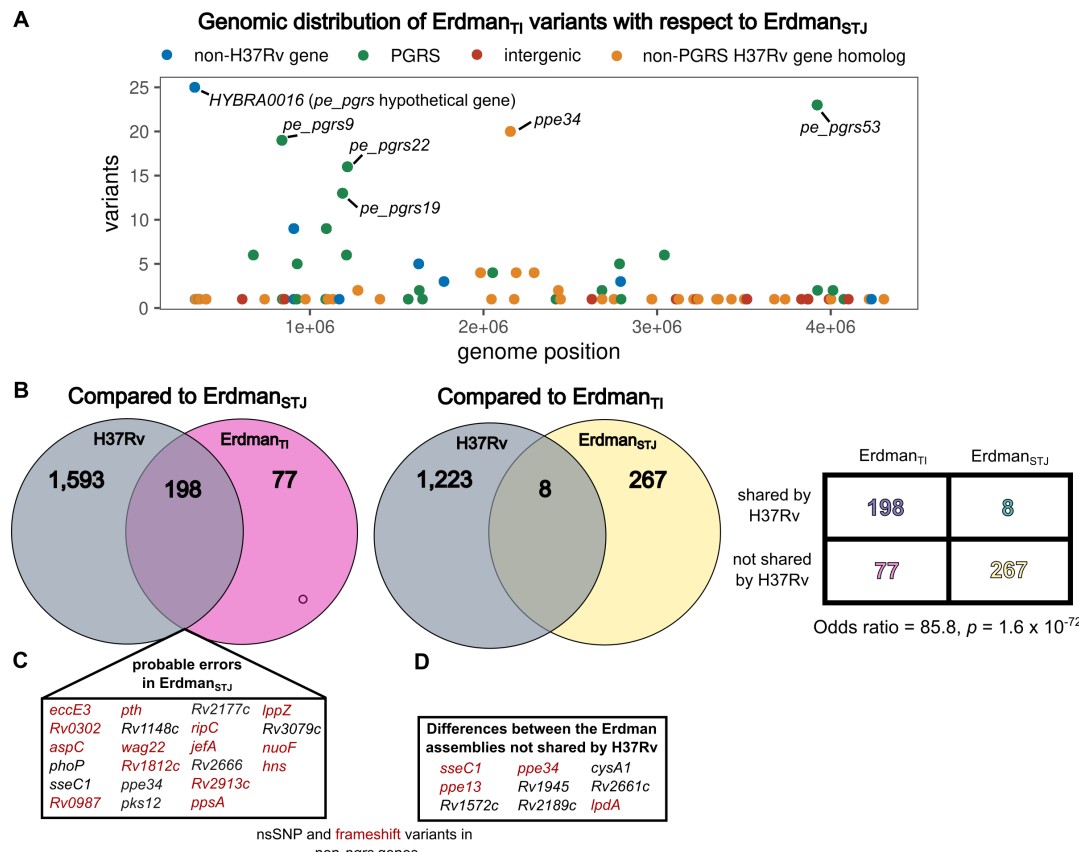

**FIG 1** Genomic differences between Erdman$_{STJ}$, Erdman$_{TI}$, and virulent type strain H37Rv. (A) Genomic distribution (*x*-axis) of and number of variants (*y*-axis) in genes in Erdman$_{TI}$ with respect to Erdman$_{STJ}$. Genes are colored by whether they belong to the PE_PGRS gene family, whether they are H37Rv homologs, and whether they are intergenic. Genes with greater than 10 variants are labeled. (B) Set comparison of genomic differences between H37Rv and the other Erdman strain with respect to Erdman$_{STJ}$ (left) and Erdman$_{TI}$ (right). Overrepresentation of H37Rv-concordant genotype in Erdman$_{TI}$ among differences between the Erdman genomes is illustrated in the contingency table on the right. All variants with respect to the Erdman genomes and their agreement with H37Rv can be found in Tables S1 and S2. (C) Set of known non-*pgrs* genes with non-synonymous or frameshift variants in both Erdman$_{TI}$ and H37Rv with respect to Erdman$_{STJ}$. (D) Similar to panel C, but for genes with variants between the Erdman assemblies that do not match H37Rv.

The set of small variants between the two Erdman genomes, where neither variant matched H37Rv (Table S1), represents either errors in one of the assemblies or genuine genomic differences evolved *in vitro*. In either case, they are important; sequencing errors highlight false variants that can mislead and produce erroneous mapping in RNA-seq and TnSeq experiments using Erdman strain derivatives, while genuine variants represent loci that vary across Erdman substrains in different laboratories. Like the differences from Erdman$_{STJ}$ common to both H37Rv and Erdman$_{TI}$, most of these variants comprised multiple mutations in a set of *pe_pgrs* genes. However, nine non-*pgrs* genes contained nsSNPs or frameshift-inducing variants (Fig. 1D) and thus represent either errors in Erdman$_{STJ}$ or loci of genuine variation among Erdman strains in different laboratories with potentially important phenotypic consequences.

## Frameshifted *lpdA* mutation prompts revisiting of the mechanism of resistance to adjunctive therapeutic candidate C10

The frameshift-inducing single-base insertion in *lpdA* (Table 1) exemplifies the importance of these variants for interpreting data from Erdman strain derivatives. Recent work tentatively attributed laboratory-evolved drug resistance to overexpression of LpdA (36). The authors hypothesized *lpdA* overexpression as a mechanism of resistance to C10, a compound that potentiates isoniazid. Erdman evolved under the administration

**TABLE 1** Probable sequencing errors in Erdman$_{STJ}$[a]

| Gene | Genome position(s) | Variants [genic position] |
|---|---|---|
| *pe_pgrs3* | 334047–334569 | 6 variants |
| *eccE3* | 357165 | 1 INS [P302fs] |
| *pe_pgrs5* | 362389 | 1 INS [G228fs] |
| *Rv0302* | 364993 | 1 INS [A6fs] |
| *aspC* | 402745 | 1 INS (G262fs) |
| 69 bp upstream *Rv0517* | 610271 | 1 SNP [T610271C] |
| 142 bp downstream *Rv0516c* | | |
| *pe_pgrs7* | 674399–674553 | 6 variants |
| *pe_pgrs9* | 838894–838927 | 19 variants |
| *pe_pgrs10* | 840529 | 1 SNP [P158S] |
| *phoP* | 851633 | 1 SNP [V35F] |
| *sseC1* | 907305–907430 | 9 variants |
| *cysA3* | 907450 | 1 SNP [T277S] |
| *pe_pgrs14* | 926594–927238 | 5 variants |
| *pe_pgrs18* | 1094652–1095270 | 8 variants |
| *Rv0987* | 1103626 | 1 INS [L678fs] |
| *pth* | 1132828 | 1 DEL [A45fs] |
| *pe_pgrs19* | 1188117–1189074 | 12 variants |
| *pe_pgrs21* | 1210869–1211444 | 6 variants |
| *pe_pgrs22* | 1215423–1216806 | 16 variants |
| *Rv1148c* | 1275205, 1275454, 1277157, 1277206 | 4 SNPs [N354H, V271L, F228F, R245R] |
| 67 bp upstream *pe_pgrs25* | 1566352 | 1 INS [T1566352TC] |
| 133 bp downstream *Rv1395* | | |
| *pe_pgrs25* | 1566451 | 1 SNP [E150Q] |
| Probable *pe/pe_pgrs* gene | 1626436–1627068 | 5 SNPs |
| *pe_pgrs28* | 1630123 | 1 INS [294/295fs] |
| | 1630345 | 1 SNP [R221G] |
| *pe_pgrs29* | 1648667 | 1 INS [294-295fs] |
| *wag22* | 1982050, 1983160 | 2 DELs [N710fs, P340fs] |
| | 1983188 | 1 MNP [G331A] |
| | 1983228 | 1 INS [A318fs] |
| *Rv1812c* | 2045871 | 1 INS [A231fs] |
| *pe_pgrs33* | 2052834, 2052910 | 2 INS [G220fs, P195fs] |
| | 2052912 | 1 DEL [A194fs] |
| *ppe34* | 2154806 | 1 SNP [H1201N] |
| *Rv1945* | 2187788, 2188202, 2188315 | 3 SNPs [L243V, I381V, G418G] |
| *pks12* | 2291402, 2291440, 2291519, 2291526 | 4 SNPs [R3887W, D3874A, R3848G, S3845S] |
| *pe_pgrs38* | 2416785 | 1 SNP [V279L] |
| *Rv2177c* | 2430971, 2431322 | 2 SNPs [P134T, N17D] |
| *ripC* | 2443997 | 1 INS [H233fs] |
| 103 bp upstream *ppe39* | 2623906 | 1 INS [G2623906GA] |
| 326 bp downstream *ppe38* | | |
| *pe_pgrs41* | 2682180, 2682209 | 2 SNPs [N152S, I162V] |
| *jefA* | 2751314 | 1 INS [T505fs] |
| *pe_pgrs42* | 2783547–2783566 | 6 variants |
| Fragmented *pe_pgrs43* | 2789199, 2789347 | 2 INS [D1542fs, R1492fs] |
| | 2789844, 2792700 | 2 DELs [D1327fs, V375fs] |
| *Rv2666* | 2970495 | 1 SNP [R181G] |
| *pe_pgrs47* | 3041667–3041711 | 6 variants |
| CRISPR repeat region (26 repeats) | 3108909 | 1 DEL [CG3108909C] |

(*Continued on next page*)

**TABLE 1** Probable sequencing errors in Erdman$_{STJ}$$^a$ (*Continued*)

| Gene | Genome position(s) | Variants [genic position] |
|---|---|---|
| *Rv2913c* | 3209370 | 1 INS [S271fs] |
| 92 bp upstream of *fpg* | 3228325 | 1 DEL [GC3228325G] |
| 268 bp downstream of *rnc* | | |
| *ppsA* | 3237065 | 1 INS [T953fs] |
| *lppZ* | 3353500 | 1 INS [P49fs] |
| *Rv3079c* | 3428082 | 1 SNP [A259P] |
| *nuoF* | 3501981 | 1 INS [F206fs] |
| *pe_pgrs53* | 3921918–3922022 | 23 variants |
| *pe_pgrs54* | 3924738 | 1 DEL [T781fs] |
| 26 bp downstream of *hsaB* | 3989739 | 1 SNP [A3989739G] |
| 260 bp upstream of *Rv3566A* | | |
| *pe_pgrs58* | 4012573, 4012880 | 2 INS [P543fs, A441fs] |
| IS1557 | 4234786 | 1 SNP [F296L] |
| *hns* | 4306113 | 1 INS [K44fs] |

$^a$Genomic differences that are common between the long-read sequenced H37Rv and Erdman$_{TI}$, with respect to ErdmanSTJ, are listed. MNP, multiple nucleotide polymorphism; SNP, single nucleotide polymorphism; DEL, deletion; INS, insertion; and fs, frameshift. Intragenic SNPs and indels are expressed as protein variants, while MNPs are expressed as genic nucleotide changes. Intergenic variants are expressed as genome position. Genes with five or more variants are collapsed. Genes with only synonymous SNPs are not listed. Full list of base-specific SNPs, MNPs, and nucleotide sequences of the deleted/inserted genomic content is in Table S1.

of C10 selected for a mutant with a single-base insertion proximally upstream of *lpdA* in the RNA polymerase binding site of its promoter. This promoter mutation increased transcription of the *lpdA-glpD2* operon four- to eightfold (36). However, the *lpdA* frameshift variant we identified in Erdman$_{TI}$ muddies this picture; a 4-bp insertion (**G**3689897**GGATC**) causes a frameshift at the 348th amino acid of LpdA. Notably, this frameshift occurs outside of the region captured by the qRT-PCR primers used by Harrison and colleagues (36) for evaluating LpdA expression changes. The frameshift is located between the FAD/NAD(P)-binding protein domain (amino acids 4–332) and the pyridine nucleotide-disulfide oxidoreductase, dimerization domain (amino acids 352–461) of LpdA. As the frameshift precedes its catalytic domain, it ostensibly abrogates LpdA oxidoreductase function.

To assess whether the *lpdA* frameshift is widespread among Erdman strains, we downloaded and evaluated whether the frameshift was present in short-read sequencing data from the C10 resistance study and a second recent study (37) (an Erdman strain derivative received by Albert Einstein College of Medicine from Trudeau Institute in the early 1990s, NCBI accession SRR23080332). All Erdman substrains contained the frameshift, suggesting either that its absence from Erdman$_{STJ}$ is erroneous, or that many circulating Erdman strains possess the frameshift (Tables S3 to S6). From the C10 study, both the parent Erdman strain (SRA accession SRX17847888) and the *lpdA* promoter mutant (GHTB146, SRA accession SRX22146880) that emerged under isoniazid ( INH) and C10 co-administration possessed the frameshift, meaning that the C10 resistance the authors observed was not due to increased oxidoreductase activity of LpdA, as it would not be translated. This implies that either the intact LpdA FAD/NAD(P)-binding protein domain or the action of GlpD2 likely underpins the phenotypic resistance to INH-C10 co-administration observed in the study.

The presence of the LpdA frameshift in our study and in the Erdman substrain from two other laboratories implies one of two scenarios. The first scenario is that the *lpdA* frameshift we observe in Erdman$_{TI}$ is a genuine variation in Erdman between labs. In this case, some strains would not have C10 resistance conferred through the *lpdA* overexpression, as it is nonfunctional, and some Erdman substrains may have enhanced susceptibility to C10, which could confound future work investigating C10 with Erdman derivative strains. In the second scenario, the *lpdA* frameshift is erroneously absent in Erdman$_{STJ}$, and the frameshift is ubiquitous in Erdman strains.

## HiFi long read-assembled genome corrects the Erdman strain genome structure

Next, we compared large sequence polymorphisms between Erdman$_{STJ}$, Erdman$_{TI}$, and H37Rv (Tables S7 and S8). Structural variants were surprisingly abundant ($n$ = 28; 5 deletions, 23 insertions; Table 2), accounting for 20,878 bp differences in total and 16,700 bp net between Erdman$_{STJ}$ and Erdman$_{TI}$—roughly 0.04% of the total genome —more than the fraction of discordant bases according to sequence identity (0.03%). This affirms the notion that SVs account for a substantial amount of sequence diversity that is systematically obscured by the predominance of short-read technologies in *Mtb* comparative genomics studies. Half of these differences (14/28, Table 2) are shared with H37Rv (11,633 bp total difference, 9,997 bp net). In contrast, H37Rv agreed with the Erdman$_{STJ}$ genome structure for only 2/28 differences in genome structure between the two Erdman assemblies (Table S8), suggesting most of them represent errors in the Erdman$_{STJ}$ genome assembly. The 28 SVs were present in 18 unique genes, and 5 of the SVs were intergenic, with SVs occurring in *pe*, *ppe,* or *pe_pgrs* genes much more frequently than other gene families (odds ratio: 141.7, *P* = 1.57e-19). The abundance of SVs in *pe/ppe* genes is consistent with previous observations of long-read sequencing resolving these repetitive, highly homologous gene families more effectively than assemblies using first- and second-generation DNA sequencing platforms (14, 38).

The greater SV concordance between H37Rv and Erdman$_{TI}$ implies that some of them represent errors in the Erdman$_{STJ}$ genome structure. However, some may be shared between H37Rv and Erdman$_{TI}$ due to common ancestry and may have legitimately evolved *in vitro* after the separation of the two sequenced Erdman samples from the original Erdman isolate. Several of the SVs not shared with H37Rv resemble genuine SVs observed to emerge between closely related laboratory strains (see "known variability" column, Table 2), suggesting some of these SVs are likely genuine. However, the stark disparity in SVs shared by H37Rv with the two Erdman strains suggests that many are indeed errors in the Erdman$_{STJ}$ assembly.

## Structural differences between H37Rv and Erdman reveal a bias toward in-frame variants in PE/PPE genes

With long-read reference genome assemblies in hand for both H37Rv and Erdman, we can now definitively provide the Regions of Difference (RDs) between the two reference strains (Table 3). In total, there are 103 differences in genome structure between H37Rv and Erdman$_{TI}$ (42 deletions, 58 insertions, 2 inversions, and 1 translocation, Table S9). Previously described RDs ($n$ = 9) and VNTRs ($n$ = 29) account for 38 of the structural differences, while IS6110 transposition events ($n$ = 16), inversions ($n$ = 2), and CRISPR region variants ($n$ = 3) account for another 21 (Table 3). One of the IS6110 insertions is upstream of gene *ppe38*, which is present in an operon with *ppe71*, *esxX,* and *esxY*. Deletion of the operon has been previously shown to make *Mtb* hypervirulent (48). Of the remaining 44 structural differences, 39 are intragenic, predominantly affecting *pe/ppe* family genes ($n$ = 29). An intriguing result of this comparison is that intragenic *pe/ppe* structural variants were overwhelmingly in-frame (17/25 as opposed to the one-third expected by chance). This in-frame bias may have been obscured to date in comparing reference genomes due to frequent misassembly of *pe/ppe* genes. Considering their predominant non-essentiality *in vitro* (49), we reason that this tendency for in-frame *pe_pgrs* indels is a result of mechanism rather than selection. This supports the idea that *pe/ppe* genes' hypermutability and frequent recombination operate as a mechanism for generating useful functional diversity, rather than one of degeneracy and gene decay.

These SVs are critical to be aware of when interpreting phenotypic differences between Erdman and other reference or clinical strains and for functional profiling studies that rely on mapping to a reference genome. *moaX* encodes a fused molybdopterin synthase with MoaD- and MoaE-like domains that requires cleavage at the Gly82 residue for MoaE-MoaD2 interaction and enzymatic function (54). A 1,052-bp deletion begins in the middle of *moaX* and encompasses *moaC3* and *Rv3324A*. Four of the nine

**TABLE 2** SVs in Erdman$_{TI}$ with respect to Erdman$_{STJ}$[a]

| Gene | SV type | Length | Start | End | Known variability |
|------|---------|--------|-------|-----|-------------------|
| **Not matching H37Rv** | | | | | |
| Probable *pe/pe_pgrs* gene | Deletion | −927 | 336007 | 336934 | |
| Probable *pe/pe_pgrs* gene | Insertion | 2,826 | 339157 | 339157 | |
| *Rv1319c* | Insertion | 1,674 | 1479546 | 1479546 | Deletion-fusion in H37Rv with respect to CDC1551 (39) |
| *ppe24* | Insertion | 150 | 1966979 | 1966979 | Hypervariable across clinical isolates (40) |
| *ppe24* | Insertion | 312 | 1967912 | 1967912 | |
| *dop* | Insertion | 57 | 2365326 | 2365326 | VNTR2372 (41) |
| 280 bp downstream of *ftsY* | Insertion | 448 | 3221354 | 3221354 | VNTR3239 (41) |
| 365 bp upstream of *amt* | | | | | |
| 116 bp downstream of *ddlA* | Insertion | 117 | 3325325 | 3325325 | VNTR3336 (41) |
| 57 bp downstream of *Rv2980* | | | | | |
| *pe_pgrs53* | Insertion | 312 | 3921669 | 3921669 | (42, 43) |
| *pe_pgrs53* | Deletion | −344 | 3921678 | 3922022 | |
| *pe_pgrs54* | Insertion | 75 | 3924176 | 3924176 | (14) |
| *pe_pgrs54* | Insertion | 36 | 3925553 | 3925553 | |
| *pe_pgrs54* | Insertion | 1,194 | 3925669 | 3925669 | |
| *Rv3611* | Insertion | 777 | 4034094 | 4034094 | (44) |
| **Matching H37Rv** | | | | | |
| *pe_pgrs7* | Insertion | 720 | 674050 | 674050 | |
| *pe_pgrs13* | Insertion | 1,332 | 924708 | 924708 | Truncated or fused with *pe_pgrs12* in several lineages (43) |
| *pe_pgrs14* | Insertion | 186 | 926583 | 926583 | |
| *pe_pgrs18* | Deletion | −422 | 1094212 | 1094634 | |
| *pe_pgrs20* | Insertion | 777 | 1190047 | 1190047 | L1-specific gene fusion (43) |
| 80 bp upstream of *pe_pgrs25* | Insertion | 1,095 | 1566339 | 1566339 | |
| 120 bp upstream *Rv1395* | | | | | |
| *pe_pgrs26* | Insertion | 57 | 1611684 | 1611684 | Frequent fs variants (multiple lineages) (45) |
| Probable *pe/pe_pgrs* gene | Insertion | 456 | 1626415 | 1626415 | |
| *pe_pgrs33* | Insertion | 315 | 2052698 | 2052698 | Large variants (multiple lineages) (46) |
| *pe_pgrs47* | Insertion | 351 | 3041653 | 3041653 | |
| 412 bp downstream of *cobB* | Deletion | −164 | 3144309 | 3144473 | QUB15 (41) |
| 29 bp upstream of *cysG*) | | | | | |
| 597 bp upstream of *Rv3304* | Deletion | −232 | 3676271 | 3676503 | VNTR3690 (2, 14, 41, 47) |
| 12 bp upstream of *lpdA*) | | | | | |
| *pe_pgrs53* | Insertion | 1,128 | 3919372 | 3919372 | (42, 43) |
| *pe_pgrs56* | Insertion | 4,398 | 3931360 | 3931360 | Fused with *pe_pgrs55* in all lineages but L5 (23, 43) |

[a]SVs are separated into those also shared by long-read sequenced H37Rv and those unique to the updated Erdman assembly. Deleted/inserted sequences can be found in Table S7. Genes with previous reports of SV among reference strains or hypervariable structure in clinical strains are cited.

non-*pe/ppe* intragenic SVs that are not known RDs or VNTRs are frameshifting (*Rv0064, Rv1928c, mamB,* and *Rv2885c*). The 5,000-bp variant in *mamB* restores its frame and reverts a truncation of the DNA methyltransferase that ablates DNA adenine methylation of its target motif, which is active in most clinical isolates (55). A 904-bp deletion disrupts *Rv0064*, a transmembrane protein implicated in response to phagosome-associated stress such as nutrient starvation (56). A 22-bp deletion frameshifts *Rv1928c*, which encodes a member of the short-chain dehydrogenase/reductase family hypothesized to play a role in the evolution of isoniazid resistance associated with redox metabolism (57). An 86-bp deletion frameshifts *Rv2885c*, which encodes a transposase of IS1539 insertion sequence and is predicted to be essential for survival *in vivo* (58). These genes maintain

**TABLE 3** SV in updated Erdman assembly with respect to HiFi-assembled H37Rv[a]

| Category | No. of SVs | No. of genes affected | Genes affected |
|---|---|---|---|
| Inversions | 2 | 4 | *Rv0794c-Rv0797* |
| Translocations | 1 | 16 | *Rv1572c-Rv1586c* (RD3 [50]) |
| Multi-gene deletions | 6 | 28 | *Rv1353c-Rv1356c* (RD145 [50]), *Rv2270-Rv2280* (RD182 [50]), Rv3018c-Rv3021c (RD514 [51]), *pe_pgrs57,* and three non-H37Rv *pe_pgrs* genes, similar to RDbovis(c)_fadD18 (52), *helZ and Rv2102* (RD178 [50]), *moaX-moaC3-Rv3324A* |
| IS6110 transposition | 16 | 15 | Inserted in Erdman_TI: *mmpS1, Rv0963c, Rv1724c, Rv1754c, Rv2336, Rv2819c, ppe38*<br>Deleted in Erdman_TI: upstream of *Rv1371, PE22, Rv2166c, Rv2478c, Rv3188, dxs2, bpoA,* and downstream of *Rv2647* |
| *pe/ppe* intragenic | 33 | 17 | In-frame: *pe_pgrs2, ppe8, ppe10* (VNTR531), *pe_pgrs6, pe_pgrs10, pe_pgrs16, pe_pgrs20, pe_pgrs21, pe_pgrs27, pe_pgrs28, ppe24* (VNTR1982), *ppe34* (VNTR2163a), *pe_pgrs38, ppe38, ppe54, pe_pgrs53, pe_pgrs61* |
|  |  | 8 | Frameshifting: *pe_pgrs3, pe_pgrs4, ppe50* (RD516_L4.1 [51]), *ppe53, pe_pgrs50, pe_pgrs54, ppe69,* non-H37Rv *pe_pgrs56* homolog |
| Non *pe/ppe* intragenic | 21 | 10 | Known VNTRs: *fhaA, Rv0071, Rv0487, rplW, Rv1435c, Rv1458c, Rv2090, aftC, Rv2680, Rv3611* |
|  |  | 2 | Known RDs: *Rv1319c* (seen in CDC1551 [39]), *lppA* (RD196 [50]) |
|  |  | 9 | Not known RDs or VNTRs: *Rv0064,[b] Rv1637c, Rv1883c, lppD, Rv1928c,[b] mamB,[b] Rv2407, Rv2885c,[b] accE5* |
| *cis*-regulatory elements | 21 | 20 | Known VNTRs: *Rv0480c, regX3, ercC3, Rv1289, Rv1290c, Rv1668c, Rv1729c, Rv1817, helY, dop, Rv2141c, qcrB, Rv2258c, amt, ftsY, Rv2980, ddlA, Rv3680, whiB4, dnaQ* |
|  |  | 5 | Not known VNTRs: *vapB1, Rv0647c, Rv3060c, Rv3327, ppe59* |
| CRISPR | 3 | 0 |  |

[a]Number of SVs in Erdman_TI with respect to H37Rv and genes affected. As the table goes down, genes associated with one category are not considered for inclusion in the next. SVs are fully detailed in Table S9.
[b]Frameshifting variants (non "*pe/ppe*" intragenic" category only). Known VNTRs retrieved from Smittipat and Palittapongarnpim (41) and Sun et al. (53).

cell wall integrity, regulate lipid metabolism, and modulate host immune responses, making them important targets for further research.

Four non-VNTR SVs affect potential *cis*-regulatory regions of non-*pe/ppe* proteins (*vapB1, Rv0647c, Rv3060c,* and *Rv3327*), all of which were insertions. Three occurred around the −10 promoter element region: a 37-bp insertion 5 bp upstream of putative antitoxin VapB1; a 230-bp insertion 13 bp upstream of probable transposase fusion protein *Rv3327;* and a 533-bp insertion 17 bp upstream of *Rv0647c*. Rv0647c encodes an essential uncharacterized gene that may be involved in mycolic acid biosynthesis (59) and lies upstream of probable lipase/esterase *lipG* and a mycolic acid synthase operon (*mmaA1, mmaA2, mmaA3,* and *mmaA4*). The other is a 68-bp insertion 36 bp upstream of the FadR subfamily GntR family transcription factor *Rv3060c*, which could in turn influence the expression of its regulon. These genes help *Mtb* withstand host immunity, maintain cell wall integrity, and adapt to nutrient limitations.

## Phase variable tandem repeats

While updating the reference genome, we also uncovered structural features that extend beyond static sequence corrections. Our analyses revealed VNTRs within subpopulations of the reference strain, suggesting that the genome is not fixed but subject to phase-variable changes. Such repeats were not only observed in the laboratory reference but also across a diverse set of clinical isolates, highlighting their potential role as dynamic elements of the *M. tuberculosis* genome. These findings provide a natural transition from updating the reference sequence toward examining how VNTR variation may contribute to phenotypic diversity, including traits linked to virulence, persistence, and host adaptation.

## A tandem duplication of the *dop* promoter may enhance nitric oxide resistance in the Erdman strain

Among the SVs specific to Erdman$_{TI}$ is a 57 bp tandem duplication of the *dop* promoter region that creates a third copy of the promoter in Erdman, two more than are present in CDC1551, and one more than in H37Rv and Erdman$_{STJ}$ (Fig. 2). While this insertion falls within the protein-coding region of Dop as annotated on Mycobrowser, the UniProt annotation (accession no. D9HNS7) suggests that it falls within the *dop* promoter, which is affirmed by transcription start site mapping (60) in H37Rv. Dop deamidates Prokaryotic Ubiquitin-like Protein (Pup) and displays depupylase activity (61). These Pup-altering activities give Dop modulatory control over the degradation of a subset of proteins that are critical for *Mtb* virulence (62). Transposon disruption of *dop* in Erdman significantly reduces fitness in antibiotic-treated stationary phase (15), suggesting that Dop also plays a clinically important role in adaptation to drug pressure.

Tandem repeat CNV is a common source of variation hotspots in bacteria due to their propensity for strand-slippage and recombination (63). Given this, it is unclear whether the *dop* promoter variant represents an error in Erdman$_{STJ}$ or is a result of genuine *in vitro* evolution. To evaluate whether the *dop* promoter CNV genuinely evolved *in vitro* in Erdman strains in different labs, we mapped reads from a recent short-read sequencing experiment on Erdman from a different laboratory (18) (BioProject PRJNA950969) to see whether they supported the two copies reported in Erdman$_{STJ}$ or the three copies in Erdman$_{TI}$. The short-read SV caller dysgu (64) identified a 57-bp insertion at the tandem repeat locus against Erdman$_{STJ}$ and did not identify any SVs at the locus

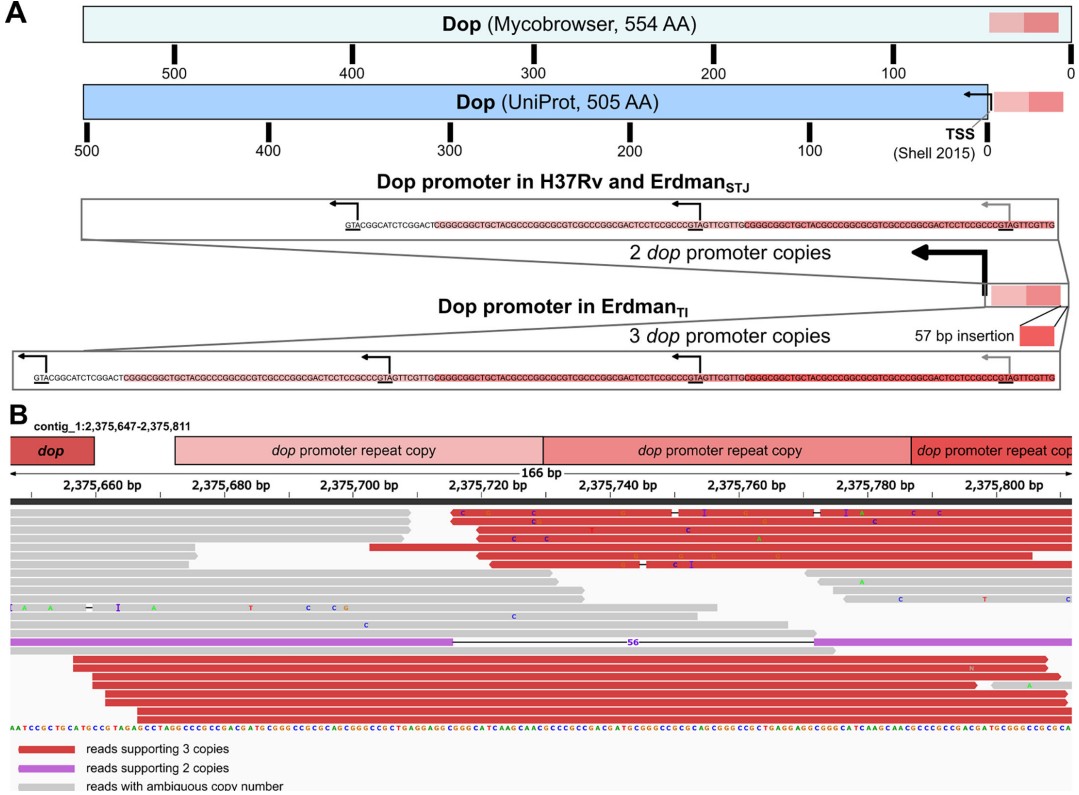

**FIG 2** Tandem repeat expansion in the promoter of Deamidase of Pup (Dop) in the Erdman strain. (A) Dop coding region as annotated by Mycobrowser (top) and UniProt (bottom), with transcription start site (TSS, arrows) annotated (60). The 57-bp element tandemly duplicated in HiFi-sequenced Erdman assembly is indicated by boxes, with sequence shown and colored by copy. Black nucleotide is the sequence comprising the rest of the region upstream of the known TSS. (B) Evidence supporting the existence of subpopulations harboring both dop promoter copy number variants. Illumina sequencing reads mapped from a sequencing experiment performed on an orthogonal Erdman strain (SRA accession SRX19884569). Reads are colored by the number of *dop* promoter tandemly repeated copies they support. The image was generated in Integrative Genomics Viewer from reads aligned to Erdman$_{TI}$.

against Erdman$_{TI}$, supporting the presence of three copies of the tandem repeat, as in Erdman$_{TI}$ (Table S10). Repeating this analysis with three additional short read data sets (SRA accessions SRX22146880, SRX17847888, and SRX19033252) from different Erdman strains revealed a variable number of *dop* PTR copies (57 and 144 bp) inserted with respect to Erdman$_{STJ}$, suggesting a hypervariable *dop* PTR copy number (Table S11 to S13). To further evaluate *dop* PTR heterogeneity, we mapped the reads to Erdman$_{TI}$ and found one read that clearly supported two copies of the tandem repeat, while the rest supported three copies (Fig. 2B). From these results, we conclude that Erdman generally has three copies, at least in some substrains, but there are subpopulations with alternative copy numbers. PTR expansion frequently occurs in bacteria (65, 66)—particularly in pathogens—and can function as a transcriptional modulator (66). PTR expansion has thus been proposed as a mechanism for phase variation, with expression often increasing as a function of tandem repeat copy number (67). This finding of hypervariable *dop* PTR copy number is important, given the influence of Dop on protein degradation and its role in *Mtb* virulence. The existence of three or more copies in most Erdman substrains may explain some of the strain's distinct manifestations in animal infection models (10, 13, 68), as the Pup proteasome curtails nitric oxide toxicity to engender pathogenesis *in vivo* (69).

### Promoter tandem repeat CNV implicates phase variable biofilm formation, drug efflux, and glycerol utilization phenotypes in M. tuberculosis

Noting this frequent variation in the *dop* promoter copy number, we searched for other instances of tandem repeat CNV in SVs between the two Erdman assemblies. This revealed a similar phenomenon for two other tandemly repeated promoter elements proximally upstream of leaderless promoters of *lpdA* and *cysG* (Fig. 3A). *lpdA* shares an operon with *glpD2*, while *cysG* has *efpA* immediately downstream and co-oriented, suggesting likely co-expression upon transcription from the *cysG* promoter. For these copy number repeats, the repeated regions are present with more copies in Erdman$_{STJ}$ than in Erdman$_{TI}$. This suggests that the CNV is genuine rather than erroneous, as errors in the Erdman$_{STJ}$ assembly would be expected to falsely collapse multiple copies into fewer than truly exist, rather than spuriously expand them.

To assess whether these *dop*, *lpdA*, and *cysG* PTR CNVs occur frequently during *Mtb* evolution, we retrieved their PTR copy number from CDC1551, Erdman, H37Rv, and a phylogeographically diverse set of recently published finished-grade long-read genome assemblies from 93 clinical *Mtb* isolates (55) and mapped them to their phylogenetic tree (Fig. 3B). This revealed that PTR copy number varies among clinical isolates for all three PTRs, even at the sublineage level, across multiple sublineages (Fig. 3B). The three PTRs each came in multiple distinct copy numbers in at least 5/15 sublineages for which we had at least two clinical isolates (Fig. 3B). These multiple PTR copy number sublineages span all four major lineages for *lpdA-glpD2* and *cysG-efpA*, and three major lineages for *dop*, suggesting their variability is a consistent feature across *Mtb* strains.

The 54–56 bp imperfect tandem repeat *cysG* promoter sequence (two copies have four Gs in a guanine homopolymer tract, while the third copy has six Gs) lies 29 bp upstream of *cysG* and has two copies fully deleted in H37Rv and Erdman$_{TI}$ with respect to Erdman$_{STJ}$, and a third copy is partially deleted (Fig. 3A).

CysG encodes a protein sequence similar to well-characterized *Salmonella typhimurium* C-2 and C-7 uroporphyrinogen-III methyltransferase (70), which converts uroporphyrinogen III to cob(II)yrinic acid a,c-diamide in the first step of the cobalamin (Vitamin B$_{12}$) biosynthesis pathway. If CysG expression changes when the copy number of its PTR increases, the resulting phenotypic effect is unclear. Current evidence suggests that *Mtb* has little or no capability of producing B12 and acquires B12 by scavenging it from the host rather than through synthesis (71). Though others have noted that some bacterial species synthesize B12 only in specific conditions that may be challenging to recapitulate *in vitro* (72). Whether CysG has a role in B12 synthesis under yet undiscovered specific environmental conditions, has some alternative function, or is simply

part of a decaying B12 biosynthetic vestige remains unclear. Irrespective of potential effects through CysG expression, the copy repeat number variation may affect the expression of its co-oriented and immediately downstream gene, *efpA* (*Rv2846c*). *efpA* is broadly conserved in *Mycobacteria* and encodes a multi-drug efflux pump that markedly decreases drug uptake and confers high levels of tolerance to first- and second-line anti-TB drugs when overexpressed in *Mycobacterium smegmatis* (73). In *Mtb*, EfpA is overexpressed in resistant clinical isolates upon exposure to first-line drugs (73) and is essential for regrowth following prolonged INH exposure (74). We speculate that phase variable expression of *efpA* provides an adaptive mechanism for constitutively generating drug-tolerant bacterial subpopulations (Fig. 3).

The 58-bp tandemly repeated *lpdA* promoter sequence lies 12 bp upstream of *lpdA* and is an exact deletion of four copies in H37Rv and Erdman$_{TI}$ with respect to Erdman$_{STJ}$ (Fig. 3A).

Given the frameshifting *lpdA* variant in Erdman$_{TI}$ described earlier and that we found to be present in other Erdman strains, LpdA overexpression would not have a clear phenotypic effect in many Erdman substrains. However, assuming the wild-type *lpdA* sequence in Erdman$_{STJ}$ is genuine (and thus LpdA is functional in some Erdman substrains), then differential LpdA expression conferred by *lpdA-glpD2* PTR copy number could influence the propensity for biofilm formation in those strains. Recent work showed that during biofilm growth, *Mtb* evolves expression-increasing *lpdA* promoter mutations, and *lpdA* is overexpressed. Strains engineered to have a second *lpdA* PTR copy produced increased biofilm biomass, demonstrating that greater *lpdA* expression enhances biofilm formation (75). In Erdman strains with nonfunctional LpdA due to a frameshift in Erdman, *lpdA-glpD2* PTR CNV would still affect expression levels of GlpD2 (Fig. 3A). GlpD2 is a glycerol-3-phosphate dehydrogenase that is likely important for glycerol utilization, is highly essential for *Mycobacterium bovis* during bovine TB infection, and has been proposed as a potential drug target for compounds aiming to disrupt the electron transport chain (76). Moreover, glycerol-phosphate metabolism was recently identified as one of the main adaptive strategies employed by multiple human bacterial pathogens during within-host evolution (77) and has previously been shown to increase antibiotic sensitivity in *Mtb* (78)—both as a fixed resistance mechanism and phase-variable drug-tolerant state (25). Thus, *lpdA-glpD2* PTR CNV may also impact the carbon substrate utilization pathway through modulating the efficiency of glycerol utilization via GlpD2 expression levels.

CNV of this *lpdA* promoter element has been observed previously between H37Ra and H37Rv, among H37Rv strains from different laboratories (2, 14) and among clinical isolates (Fig. 3B). This suggests that it evolves frequently and is a likely source of phase variation, creating subpopulations with differential propensity for biofilm formation via LpdA expression levels and glycerol utilization via GlpD2 expression levels (Fig. 3A). Given their co-operonic configuration, we speculate the LpdA promoter serves as a phase-variable mechanism for biofilm formation and glycerol utilization during biofilm growth. This would also explain the related observation of especially high *lpdA* copy number in several laboratory strains, which use glycerol as a carbon source.

### Expansion and contraction of tandem repeats across clinical strains are paralleled in subpopulations

PTR expansion frequently occurs in bacteria (65, 66)—particularly in pathogens—suggesting it can function as a transcriptional modulator (66). PTR expansion has thus been proposed as a mechanism for phase variation, with expression often increasing as a function of tandem repeat copy number (67). To test the hypothesis that these variable PTR regions are phase variable, we investigated whether the hypervariable PTRs were emerging frequently enough to appear within the subpopulations sequenced for Erdman$_{TI}$. We leveraged the ultra-deep HiFi sequencing data to scan for individual deletions or insertions of PTR copies (Fig. 4). Three sequencing reads identified subpopulations with fewer *dop* promoter copies than the consensus sequence; two

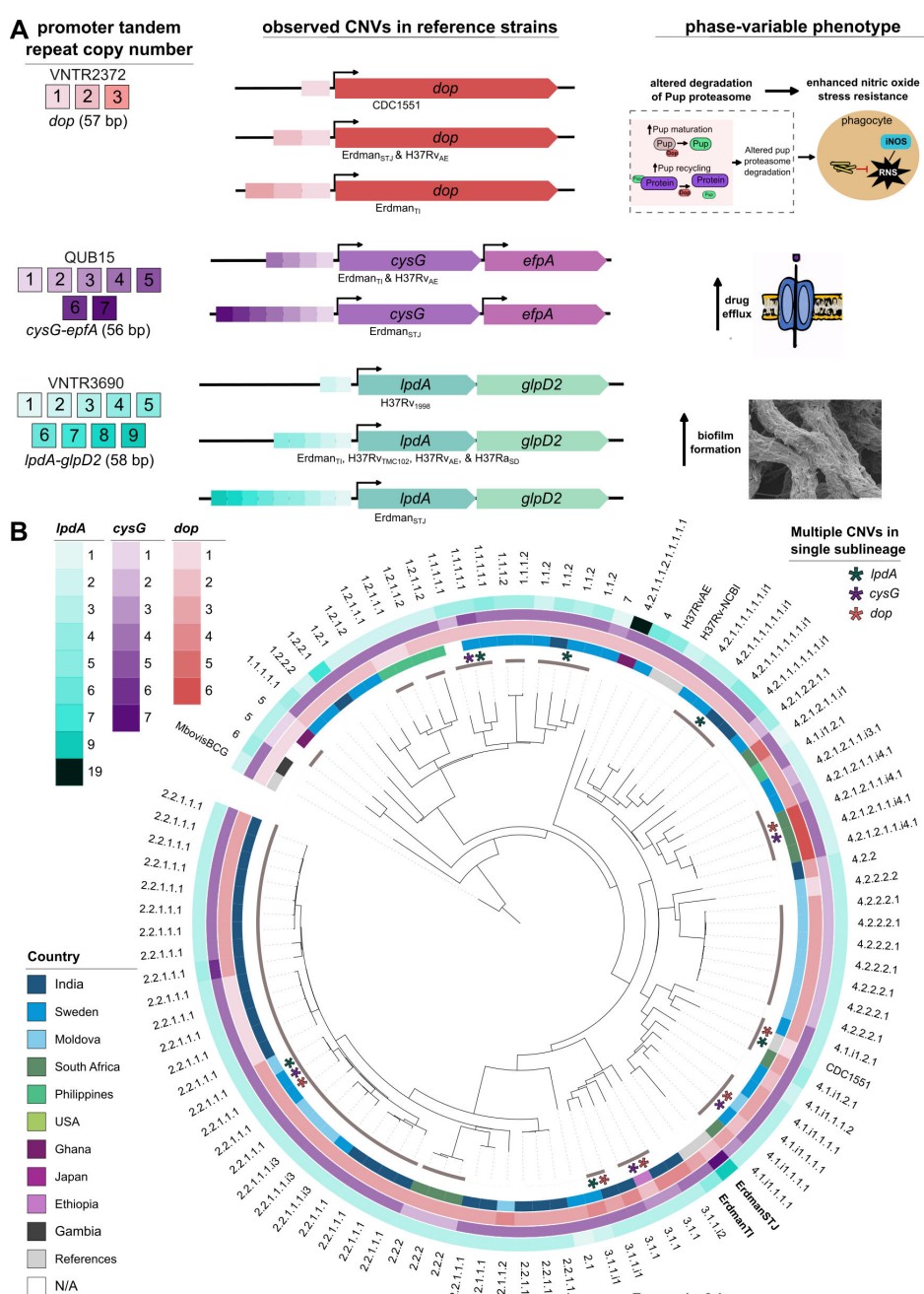

**FIG 3** Frequent promoter CNV as a mechanism for phase variation of clinically relevant phenotypes in *M. tuberculosis.* (A) Promoter copy number and length of the tandemly repeated promoter sequence (left column) and their variation in primary *Mtb* laboratory strains (middle column). VNTR IDs are noted for each tandem repeat. Potential phenotypic effects of differential expression due to increased promoter activity are depicted in the right column. (B) Phylogenetic tree colored by the copy number of 93 clinical isolates, finished-grade *de novo* assembled and published previously (55), along with CDC1551 and H37Rv. Sublineage is annotated for each isolate, with tips colored according to the number of copies of the repeated promoter sequence element. Erdman genomes are bolded. Sublineages with multiple isolates are outlined (gray inner track) and have asterisks when multiple copy numbers are observed within the sublineage. Asterisks are colored according to promoter tandem repeats with multiple copy numbers within the sublineage.

reads harbored deletion of two *dop* PTR copies, and one harbored deletion of a single copy (Fig. 4A). Similarly, one sequencing read identified a subpopulation with a single *lpdA-glpD2* PTR copy deleted (Fig. 4B). No reads supporting the existence of alternative

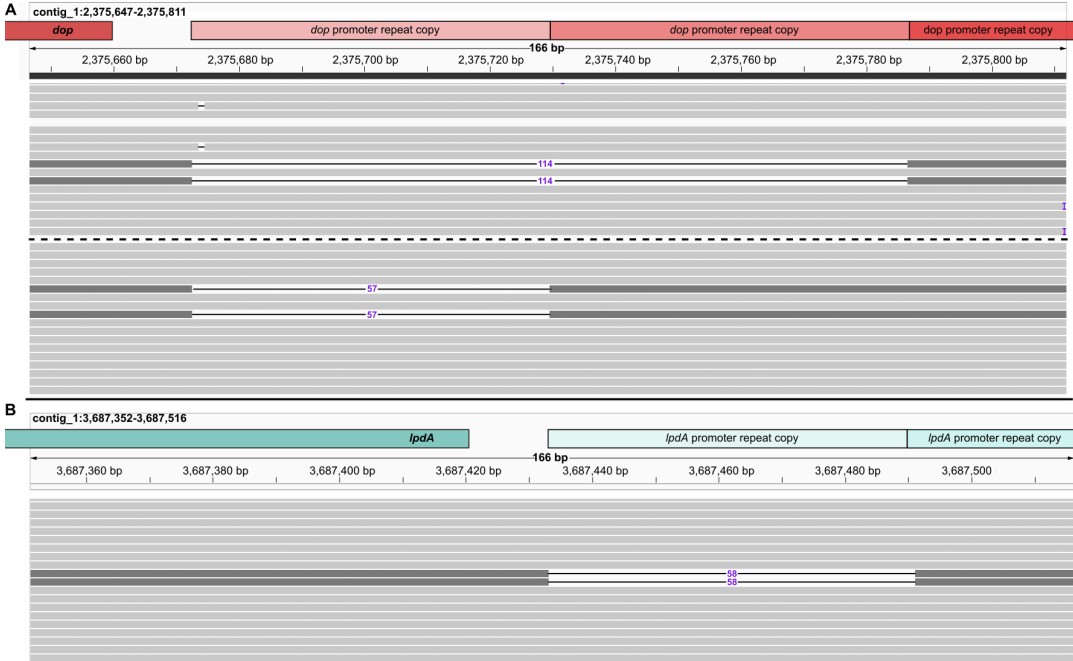

**FIG 4** Evidence of phase variation of *dop* and *lpdA-glpD2* operons through high-frequency promoter CNV in *M. tuberculosis*. Images of individual HiFi sequencing reads from the Erdman strain demonstrating subpopulation with copy number variants of tandemly repeated *dop* (A) and *lpdA* (B) promoter sequences. HiFi reads are aligned to the Erdman<sub>TI</sub> consensus sequence with CNV-supporting reads highlighted (dark gray). Length of the deleted sequence with respect to the consensus genome is shown in the middle of the thin black line connecting the aligned portions of the read. Tandemly repeated promoter copies and coding sequences are shown at the top. Visualizations were generated with Integrative Genomics Viewer (80).

copy number PTRs were observed for the *cysG-efpA* PTR. This, however, certainly does not preclude their existence. Even at the exceptionally high depth we sequenced the Erdman strain, we would still only expect to detect PTR copy number variants on the order of 1 in 1,000 to 1 in 10,000—a much higher frequency than would be required to reliably generate alternative phases during infection, when bacterial numbers easily exceed 1 billion (79).

Early *Mtb* comparative genomics studies noted the existence of several dozen loci in the genome where tandemly repeated sequences often varied among laboratory and clinical strains (28). Excitement over these regions was primarily in their utility for lineage typing, as copy number variants often became fixed and, particularly in combination, enabled the development of barcodes that were state-of-the-art at the time for quickly and affordably discriminating between various *Mtb* lineages and for defining outbreaks (29). As RDs were discovered and SNP typing became more widely available, VNTR typing fell out of favor. With a few notable exceptions, the study of VNTR regions was largely cast aside, owing to their primary use for lineage typing, and from the hypothesis that VNTR elements were "selfish" genetic elements (81), which was gaining a foothold in the literature.

On the contrary, bacteria wielding highly reversible genetic variations as a mechanism to continuously instantiate subpopulations with different phenotypes than the majority is a widespread phenomenon, known as "phase variation," with many well-characterized examples (82). Phase variation is particularly prevalent in pathogens (83, 84). In *M. tuberculosis*, phase variation through ON/OFF gene function conferred by homopolymer tract length variation (27) and *cis*-regulatory effects of mosaic DNA methylation patterns (55) have recently been identified as previously overlooked mechanisms of phenotypic plasticity. A particular example of an INH-resistant mutant has already demonstrated the potential phenotypic impact and clinical relevance of such repeat expansion (85). We contend that tandem repeat copy number expansion/contraction

represents a third central mechanism of phase variation in *Mtb*. The three PTRs described in our analysis each fall in promoters whose varied expression would engender subpopulations with phenotypic consequences advantageous in some of the diverse microniches that *Mtb* inhabits during infection. There remain dozens of other known regions with similarly variable *cis*-regulatory regions. A particularly incisive example is *lpdA*, as it is one of the few VNTRs whose copy number has been empirically demonstrated to influence its expression; four-copy LpdA PTR exhibited a 12.5-fold increase in LpdA expression compared to one copy (47).

Phenotypic adaptation conferred by PTR copy number variants poses challenges to genetic adaptation studies that have been obscured by short-read sequencing. Experimental evolution under exposure to candidate and on-market drugs is often conducted to identify resistance-conferring mutations that emerge under drug exposure (86–89). To date, these studies have been performed nearly exclusively with short reads, which cannot resolve most VNTR copy number variants, as the sum of their copies exceeds the read length of short-read sequencing technologies. This makes it impossible to detect copy number variants where the sum of the tandemly repeated copies exceeds 150–300 bp (depending on the library preparation and sequencing platform), which most VNTRs exceed. This presents a scenario where any selective advantage conferred by copy number variants goes unnoticed, and where mutations that coincide with adaptive copy number variants are spuriously attributed as causal to the adapted phenotype. It remains to be determined how frequently this occurs, but the functions of genes downstream of many VNTRs and their tendency to coincide with *cis*-regulatory regions suggest that the frequency is well above zero. While *lpdA* is among the only VNTRs with demonstrative evidence of expression level dictated by VNTR copy number, the mechanistic rationale for others behaving similarly is strong. As the promoter region increases in copy number, so does the opportunity for effectors of transcription to bind and promote initiation of RNA polymerase open complex formation (90), potentially increasing the frequency of transcription initiation.

## Future directions

Our findings foreshadow future discoveries for contributions of VNTR copy number variants and other forms of CNV as long-read sequencing continues to grow in popularity and affordability. Targeted strategies for monitoring the emergence of tandem repeat copy number variants and how their composition varies under various selective pressures are another important future line of research, particularly for copy number variants that emerge at lower frequencies but still generate viable, phenotypically distinct phases. Wet lab experiments isolating strains with different numbers of tandem repeats from the subpopulation will be helpful to uncover the phenotypic properties these VNTRs influence. Perhaps counterintuitively, genes downstream of *cis*-regulatory VNTRs may be prudent targets for drug development. Proteins with phase-variable expression are often critical for functions essential in the bacterial life cycle, especially for pathogens, suggesting their targeting could inhibit processes essential to pathogenesis.

## Concluding remarks

Here, we HiFi-sequenced and assembled the genome of the *Mtb* Erdman reference laboratory strain and demonstrated that it provides a significantly improved reference genome for the frequently studied Erdman strain (BioProject No. PRJNA1250540). Our findings demonstrate the value of HiFi resequencing of bacterial reference strains assembled with first- and second-generation sequencing technologies for materially improved reference genomes. Doing so for the Erdman reference significantly improves genome resolution for mapping TnSeq and transcriptomic sequencing data in future studies.

Comparison of the Erdman assemblies also revealed variants unlikely to be due to sequencing errors, with several instances of tandemly repeated promoter sequences varying in copy number immediately upstream of transcription units with clear links

to phenotypes implicated in pathogenesis (Fig. 2 to 4). While tandem repeat CNV has been observed previously in *Mtb* (2, 14, 44), the frequency of these variants in laboratory strains and clinical isolates is notably high relative to small variants, which are canonically considered to be the predominant mutational force driving *Mtb* adaptivity. This suggests that tandem repeat CNV accounts for more phenotypic plasticity in *Mtb* than previously appreciated and represents a form of contingency loci for the species, providing a repertoire of alternate phenotypes that often arise in subpopulations. This finding should compel greater scrutiny of these regions in clinical isolates and lab-evolved mutants, particularly in cases where drug resistance or other phenotypes of interest are unexplained by known causal variants. More broadly, these findings add to the mounting evidence (91, 92) that CNV is a principal driver of rapid adaptation across kingdoms of life.

## MATERIALS AND METHODS

### Bacterial growth and DNA extraction

A 10 mL culture of *Mycobacterium tuberculosis* Erdman (TMC 107) bearing a chromosomally integrated copy of plasmid L5 attB::Pleft* mScarlet, a gift from Clifton Barry III (Addgene plasmid # 169410; http://n2t.net/addgene:169410; RRID:Addgene_169410) (93), was grown in 7H9/OADC/0.5% glycerol/0.05% tyloxapol media in a 30 mL square PETG bottle shaking at 37°C to an $OD_{600}$ of 0.4. This strain was prepared from a low-passage stock from the TMCC, which was maintained and passaged as pellicles on the surface of liquid media (94), as detergent-dispersed culture in liquid media is associated with mutation leading to loss of virulence, as is commonly observed for PDIM (37). The strain was not passaged after isolation before being grown for DNA extraction. Genomic DNA was isolated from the cell pellet using the Quick-DNA Fungal/Bacterial Miniprep Kit (Zymo Research) according to the manufacturer's instructions (bead-beating lysis followed by spin-column purification). The chromosomally integrated Pleft*–mScarlet cassette was included to enable a separate subsequent project and was not otherwise used in the present study; for continuity, we proceeded with the same strain here.

### DNA sequencing

DNA sequencing was performed at the Brigham Young University DNA Sequencing Center in Provo, UT, USA. DNA libraries for PacBio (Pacific Biosciences, Menlo Park, CA, USA) were prepared using PacBio's DNA Template Prep Kit with no follow-up PCR amplification. Briefly, sheared DNA was end repaired, and hairpin adapters were ligated using T4 DNA ligase. Incompletely formed SMRTbell templates were degraded with a combination of Exonuclease III and Exonuclease VII. The resulting DNA templates were purified using SPRI magnetic beads (AMPure, Agencourt Bioscience, Beverly, MA, USA) and annealed to a twofold molar excess of a sequencing primer that specifically bound to the single-stranded loop region of the hairpin adapters. SMRTbell templates were subjected to circular consensus SMRT sequencing using an engineered phi29 DNA polymerase on the PacBio Revio system according to the manufacturer's protocol.

Publicly available short-read sequencing data sets were used for comparison in the study. The source, sequencing technology, and relevant analyses are summarized in Table 4.

### Genome assembly and quality control

Raw Pacific Biosciences SMRT-sequencing reads were assembled using an in-house assembly pipeline. Sequencing reads were filtered using Filtlong version 0.2.1 (minimum read length = 1,000 bp; read quality cutoff: top 95% of reads), assembled into a single contiguous sequence, polished via the long-read assembler flye (95) version 2.9.2, and reoriented with *dnaA* as the starting gene using dnaapler (96) version 0.7.0. Flye was run with "–asm-coverage 100" so that the longest reads were used to assemble the

**TABLE 4** Sequencing data used in the study

| Description | Institution | Technology | BioProject | Run accession | Read accession | Analysis | Original source |
|---|---|---|---|---|---|---|---|
| Ultra-deep HiFi sequencing of Erdman (Erdman$_{TI}$) | San Diego State University and Trudeau Institute | PacBio Revio HiFi | PRJNA1250540 | SRX28390226 | SRR33126813 | Genome assembly | This study |
| RS II sequencing of H37Rv (H37Rv$_{AE}$) | The Wellcome Trust Sanger Institute | PacBio SMRT RSII P6C4 | PRJEB8783 | ERX2597852 | ERR2580787 | Consensus genome comparison | |
| Original Erdman genome assembly (Erdman$_{STJ}$) | National Center for Global Health and Medicine (Shinjuku, Tokyo, Japan) | Roche (GS FLX Titanium sequencer), Illumina (Genome Analyzer IIx), and ABI 3730xl | PRJDB66 | SAMD00061050 | NA$^a$ | Consensus genome comparison | 22,535,945 (9) |
| Erdman str. mutant selected under INH and C10 | Washington University | Illumina NovaSeq 6000 | PRJNA1030020 | SRX22146880 | SRR26442322 | *lpdA* frameshift analysis; *dop* promoter copy number analysis | 38,294,237 (36) |
| Parent Erdman str. mutant | Washington University | Illumina NovaSeq 6000 | PRJNA889365 | SRX17847888 | SRR21859821 | *lpdA* frameshift analysis; *dop* promoter copy number analysis | 38,294,237 (36) |
| PDIM mutant screening | Albert Einstein College of Medicine | Illumina MiSeq | PRJNA923717 | SRX19033252 | SRR23080332 | *lpdA* frameshift analysis; *dop* promoter copy number analysis | 38,740,932 (37) |
| Phenogenomic analysis of clinical strains | Harvard University | Illumina NovaSeq 6000 | PRJNA950969 | SRX19884569 | SRR24083712 | *lpdA* frameshift analysis; *dop* promoter copy number analysis | 38,734,030 (18) |

$^a$NA, not available; reads for the original Erdman genome submissions are unavailable.

initial scaffold at 100× depth, and all reads were used at later stages to complete the assembly. Genome structure quality assurance was performed by mapping reads against the consensus genome and calling structural variants with Sniffles2 (97) to ensure no majority subpopulations were called (which would indicate misassembly). No such SVs were present in the Erdman$_{TI}$ consensus genome assembly. The mScarlet construct was removed from the consensus sequence, *in silico*.

## Genome annotation

The polished, re-oriented assembly was annotated using hybran version 1.8, a hybrid reference transfer and *ab initio* genome annotation tool that utilizes a reference GenBank file to annotate genomes (98). The genomes of PacBio sequenced Erdman and the Erdman$_{STJ}$ strain were annotated using H37Rv$_{AE}$ as the reference.

## Comparative genomics

### Small variants

The Erdman reference genome (Erdman$_{STJ}$) for comparison was downloaded from NCBI (GenBank accession no. AP012340.1, RefSeq accession NC_020559.1). Genomes of HiFi-assembled Erdman (Erdman$_{TI}$), the current Erdman reference genome, and long-read-sequenced H37Rv from Albert Einstein College of Medicine (H37Rv$_{AE}$, was *de novo* assembled as described previously [55] from PacBio RSII with polymerase and chemistry P6C4 sequencing data [99] deposited in NCBI BioProject PRJEB8783, referred to as simply "H37Rv" throughout the Results section for simplicity) were compared by running Mummer dnadiff version 1.3 (100). The Mummer dnadiff output was then converted to vcf by a custom script and normalized using bcftools norm and variants annotated with vcf-annotator version 0.7 (https://github.com/rpetit3/vcf-anno-tator). Differences shared by Erdman$_{TI}$ and in H37Rv$_{AE}$ with respect to Erdman$_{STJ}$ were separated into unique differences and common differences by comparing start position and the type of SV.

Short read variant calling was performed by mapping short reads to Erdman$_{STJ}$ and Erdman$_{TI}$. Variant calling was performed using "samtools mpileup" and running VarScan (101) (version 2.4.4 flags: --min-avg-qual 10 --min-coverage 10 --variants --output-vcf 1 --strand-filter 0).

SVs were called from long-read sequencing data using svim-asm (102) (version 1.0.3). svim-asm reports five types of SVs: deletions, insertions, tandem, interspersed duplications, and inversions by using the alignment of two genomes. svim-asm analyzes genome-genome alignments in BAM format and searches discordant alignments to extract SV signatures between the two sequences. It defines SV signatures as pieces of evidence pointing to the presence of an SV between the query genome assembly and the reference assembly. Heterogeneous SVs were called by mapping reads against the consensus genome and running sniffles2 (97) "mosaic" mode with minimum allele frequency set at "0.00001."

Short-read SV callers, Delly (103) and dysgu (64) (version 1.6.2, default settings except flag "–*diploid False*)," were used to disambiguate whether a third, Illumina-sequenced sample of Erdman from a recent publication (18) had two or three copies of the *dop* promoter tandem repeat (DPTR) illustrated in Fig. 2. Short-read sequencing data were downloaded from NCBI (SRR24083712, SRX22146880, SRX17847888, SRX19033252, and SRX19884569). Both SV callers were run separately with both Erdman$_{TI}$ and Erdman$_{STJ}$ as reference and did not call any DPTR copy number SVs with either Erdman genome as reference. Delly returned no SVs with respect to either Erdman assembly, so we used results from dysgu for the analysis.

## Copy number variant subpopulation analysis

### HiFi sequencing data

For the 15,000 bp flanking each of the PTR regions (upstream of *dop*, *lpdA*, and *cysG,* Fig. 3), HiFi reads from Erdman$_{TI}$ were extracted from aligned, sorted, and indexed BAM files for Erdman$_{TI}$ to create a BAM file of just the copy number variant-containing region and the sequence surrounding it that HiFi reads would span. Each of the three PTR BAM files was loaded into Integrative Genomics Viewer (80) (IGV) and screened for insertions and deletions corresponding to multiples of the repeated sequence length in order to identify subpopulations of copy number variants for each of the three PTRs. SVG images were extracted from the IGV of reads supporting subpopulations with non-consensus copy number for visualization in manuscript figures.

### Illumina sequencing data

Illumina sequencing data (Table 2) were aligned to Erdman$_{TI}$ and Erdman$_{STJ}$, and screenshots of reads supporting the existence of each copy number were captured from IGV for figure generation (Fig. 2).

## Phylogenetic reconstruction

The phylogenetic tree was reconstructed using a concatenated list of variants from 98 complete *Mtb* genomes, including Erdman$_{TI}$, a subset of a previously reported reference set of *Mtb* clinical isolates (104) that were long-read sequenced (105) and *de novo* assembled (55), a set of clinical isolates phylogeographically diverse, finished-grade *de novo* assembled and published previously (55), as well as the H37Rv (NC_000962.3), Erdman (NC_020559.1), and CDC1551 (NC_002755.2) *Mtb* reference genomes downloaded from NCBI. *Mycobacterium bovis* BCG (AM408590.1) was used as the outgroup for this phylogeny. The list of variants was obtained by aligning the whole genome of each isolate against the H37Rv reference genome using the MUMmer dnadiff (100) version 3.23. The dnadiff outputs were converted to VCF format using a custom script, including a normalization step with "bcftools norm." The normalized VCF files were then merged into a single file using a custom script. The substitution model was calculated with ModelTest-NG version 0.2.0 (106). The maximum likelihood tree was reconstructed with

RAxML-NG (107) version 1.2.2, using 25,839 sites, the TIM1 model, and 100 bootstrap replicates. Visualization and editing of the phylogenetic tree were achieved with iTOL (108) software version 6.

## Literature review

To determine whether previous publications had falsely attributed phenotypic traits of Erdman to genotypic sources that we determined to be sequencing errors, we performed an exhaustive literature review. Using Google Scholar and the PubMed Database available through the National Institutes of Health, we searched for relevant publications using the keywords "Erdman," "H37Rv," *Mycobacterium tuberculosis,*" "comparison," and "genome." We then examined all publications directly comparing Erdman and H37Rv to determine if a genomic source had been previously linked to phenotypic differences observed between the two strains. If a genomic source had been identified, we compared those findings to the set of $Erdman_{STJ}$ genomic differences in common between H37Rv and $Erdman_{TI}$.

## ACKNOWLEDGMENTS

We thank Afif Elghraoui for assistance in processing the primary HiFi sequencing data and guidance in developing the $Erdman_{TI}$ assembly, Theron James for assistance in the Phylogenetic Reconstruction pipeline and genome annotation with hybran, and Derek Conkle-Gutierrez, Mahshid Fallahpour, and Tristan Jang for critical review and feedback on the manuscript.

This work was funded through grants (R01AI105185 and R01AI163202) by the National Institute for Allergy and Infectious Diseases (NIAID) awarded to F.V. The funding bodies had no role in the design of the study or in the collection, analysis, and interpretation of data or in the writing of the manuscript.

S.J.M., F.V., and B.C.W. conceptualized the study. N.T., S.J.M., R.L.L., B.W., and F.V. curated the data. N.T., S.J.M., and P.M.M.-P. performed formal analysis. F.V. acquired funding and contributed to project administration. S.J.M., N.T., F.V., B.W., G.G.M., P.M.M.-P., and R.L.L. performed the investigation. F.V. and B.W. provided resources. N.T., S.J.M., and P.M.M.-P. helped with software. S.J.M., F.V., and B.W. supervised the study. S.J.M., P.M.M.-P., and N.T. visualized the study. S.J.M., N.T., P.M.M.-P., B.W., R.L.L., and F.V. wrote the original draft and reviewed and edited the manuscript. All authors reviewed and approved the final manuscript.

## AUTHOR AFFILIATIONS

[1]Laboratory for Pathogenesis of Clinical Drug Resistance and Persistence, School of Public Health, San Diego State University, San Diego, California, USA
[2]Trudeau Institute, Saranac Lake, New York, USA

## AUTHOR ORCIDs

Samuel J. Modlin  http://orcid.org/0000-0002-7674-305X
Nachiket Thosar  http://orcid.org/0000-0002-5069-0236
Brian Weinrick  http://orcid.org/0000-0003-0880-4487
Faramarz Valafar  http://orcid.org/0000-0002-3648-9384

## FUNDING

| Funder | Grant(s) | Author(s) |
| --- | --- | --- |
| National Institute of Allergy and Infectious Diseases | R01AI163202 | Paulina M. Mejía-Ponce |
| National Institute of Allergy and Infectious Diseases | R01AI105185 | Gaelle Guiewi Makafe |

## AUTHOR CONTRIBUTIONS

Samuel J. Modlin, Conceptualization, Data curation, Formal analysis, Investigation, Methodology, Software, Supervision, Validation, Visualization, Writing – original draft, Writing – review and editing | Nachiket Thosar, Data curation, Formal analysis, Investigation, Software, Visualization, Writing – original draft, Writing – review and editing | Paulina M. Mejía-Ponce, Formal analysis, Investigation, Software, Visualization, Writing – original draft, Writing – review and editing | Raegan L. Lunceford, Data curation, Investigation, Writing – original draft, Writing – review and editing | Gaelle Guiewi Makafe, Investigation, Writing – review and editing | Brian Weinrick, Conceptualization, Data curation, Investigation, Resources, Supervision, Writing – original draft, Writing – review and editing | Faramarz Valafar, Conceptualization, Funding acquisition, Investigation, Methodology, Project administration, Resources, Supervision, Validation, Writing – original draft, Writing – review and editing

## DATA AVAILABILITY

All codes are available on GitLab at https://gitlab.com/LPCDRP/erdman-update. The corrected Erdman strain reference genome assembly and the raw reads are deposited in PRJNA1250540. All other data in the article were accessed via publicly available resources, as described in Table 4. This study did not generate new materials.

## ADDITIONAL FILES

The following material is available online.

### Supplemental Material

**Table S1 (mSystems01026-25-s0001.xlsx).** Small variants in ErdmanTI with respect to ErdmanSTJ with functional effects annotated.
**Table S2 (mSystems01026-25-s0002.xlsx).** Small variants in ErdmanSTJ with respect to ErdmanTI with functional effects annotated.
**Table S3 (mSystems01026-25-s0003.xlsx).** Small variants found in short read Erdman from Harvard University (SRR24083712) with respect to ErdmanSTJ.
**Table S4 (mSystems01026-25-s0004.xlsx).** Small variants found in short read Erdman from Albert Einstein College of Medicine (SRR23080332) with respect to ErdmanSTJ.
**Table S5 (mSystems01026-25-s0005.xlsx).** Small variants found in short read wild-type Erdman from Washington University (SRR21859821) with respect to ErdmanSTJ.
**Table S6 (mSystems01026-25-s0006.xlsx).** Small variants found in short read Erdman GHTB146 from Washington University (SRR26442322) with respect to ErdmanSTJ.
**Table S7 (mSystems01026-25-s0007.csv).** Structural variants in ErdmanTI with respect to ErdmanSTJ with affected genes annotated.
**Table S8 (mSystems01026-25-s0008.xlsx).** Structural variants in ErdmanSTJ with respect to ErdmanTI with affected genes annotated.
**Table S9 (mSystems01026-25-s0009.csv).** Structural variants in ErdmanTI with respect to H37Rv with affected genes annotated.
**Supplemental Tables (mSystems01026-25-s0010.xlsx).** Tables S10-S13.

### Open Peer Review

**PEER REVIEW HISTORY (review-history.pdf).** An accounting of the reviewer comments and feedback.

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
