## [Reviewer comments · mSystems]

Updated Erdman reveals tandem repeat copy number is phase-variable and impacts *M. tuberculosis* adaptation across evolutionary timescales

Samuel Modlin, Nachiket Thosar, Paulina Mejía-Ponce, Raegan Lunceford, Gaelle Guiewi Makafe, Brian Weinrick, and Faramarz Valafar

Corresponding Author(s): Faramarz Valafar, San Diego State University

Review Timeline:

Submission Date:	July 9, 2025
Editorial Decision:	August 12, 2025
Revision Received:	November 10, 2025
Accepted:	November 18, 2025

Editor: Zoe Dyson

Reviewer(s): Disclosure of reviewer identity is with reference to reviewer comments included in decision letter(s). The following individuals involved in review of your submission have agreed to reveal their identity: William R Jacobs (Reviewer #1); Conor Meehan (Reviewer #2)

Transaction Report:

DOI: <https://doi.org/10.1128/msystems.01026-25>

Re: mSystems01026-25 (**Tandem repeat copy number is phase-variable and impacts *M. tuberculosis* adaptation across evolutionary timescales**)

Dear Dr. Faramarz Valafar:

Revision Guidelines

Sincerely,
Zoe Dyson
Editor
mSystems

Reviewer #1 (Comments for the Author):

This is a very interesting study that uses long read DNA sequences to assemble Mtb genomes and has identified polymorphisms that reflect differences between the sequenced Erdman strain of Mtb and the strain that came from the Trudeau collection. I love the discussion of the results as well as the methodology, but I do suggest the following revisions:

1. Put in extreme detail how you grew the strains from the Trudeau collection.

2. Point out the George Kubica paper from 1956 that describes how the strains were stored. Its important to know that all strains were grown on solid media as opposed to liquid media, this is an important point that is often ignored by TB researchers. For example, you should reference Larry Schlesinger has always grown on 7H11 plates and that's what he put into human macrophages for his virulence studies. As I have reviewed, many of the virulence studies done in the early days of TB biology, the strains were always grown on solid media.
3. You did not mention if the Erdman strain has *esxX* or *esxY*, which is missing from the original H37Rv genome sequence. I think you should reference the recent paper from Koleske et al 2025 (Science Advances paper). This region, when deleted, makes the strain hypervirulent compared to the Erdman. Due to the PPE71 duplication it was missed when Stewart Cole's teams sequenced the ordered library of cosmids. Your technique of long reads is exciting and will become the standard of what others will follow.
4. You should mention in the Kubica paper as well as the recent Mulholland paper that passage of strains in liquid media can lead to loss of virulence. In H37Rv strains, this is due to a loss of PDIM that can occur by mutations in many different genes. The Mulholland paper should be referenced and discussed with respect to your work. It is interesting that with the Erdman strain, you do not see any mutations that result in PDIM loss. This is interesting but not understood.
5. Please reference Dr. Erdman's original work of the isolation of this Mtb strain in your paper.

Reviewer #2 (Comments for the Author):

The manuscript by Modlin et al outlines a new reference genome for the Erdman reference genome for Mtb and some nice proposals of evolutionary trajectories that are observed from the related analysis.

Overall, the paper is robust with a lot of information. I don't have many major concerns. My main issue is the 'story' that runs through. For example, the title is focussed on the tandem repeat aspect, which is actually not a focus of the majority of the paper. Indeed in the abstract and in many places in the results, it is difficult to find out which 'story' is the lead here: is it the reference genome issue that are corrected or the proposal of evolutionary paths that come from that? If the former, this is well supported and outlined in the paper. If the latter, much of this is proposed phenotypic outputs but without accompanying experimental data. I still think it is well supported from the genomic analysis, but it is not confirmed, which would require much more work.

In line with this, the paper is very dense and difficult to follow. This is my area of speciality so I understood what was being explained but for someone with less expertise, this paper would be difficult to understand in many places. The combination of results and discussion make it hard to see what is a direct result of this work and what is a proposed explanation for some of the genomic features observed (but not validated).

Generally, the paper is a strong one with a lot of relevant information. I would suggest for readability it needs to be looked at to separate out the primary findings (and ensure they are clear in the title and abstract, which they currently are not) from the proposals for future investigation. This should include some simplification of text in the results/discussion.

Minor points:

Line 460: A very short explanation of why the plasmid insertion is used here would be appreciated

Line 490: I think this is an incomplete sentence. This is an example of a few places in the text where there are grammatical/sentence structure issues that need another pass.

We thank the reviewers for the careful review and for the opportunity to increase the work's rigor, clarity, and insight.

REVIEWER #1 (COMMENTS FOR THE AUTHOR):

This is a very interesting study that uses long read DNA sequences to assemble Mtb genomes and has identified polymorphisms that reflect differences between the sequenced Erdman strain of Mtb and the strain that came from the Trudeau collection. I love the discussion of the results as well as the methodology, but I do suggest the following revisions:

Q1. Put in extreme detail how you grew the strains from the Trudeau collection.

Reply: We have added a details of strain passage at line number 468. This strain was prepared from a low passage stock from the TMCC, which was maintained and passaged as pellicles on the surface of liquid media, as detergent dispersed culture in liquid media is associated with mutation leading to loss of virulence. The strain was not passaged after isolation before being grown for DNA extraction.

Q2. Point out the George Kubica paper from 1956 that describes how the strains were stored. It's important to know that all strains were grown on solid media as opposed to liquid media, this is an important point that is often ignored by TB researchers. For example, you should reference Larry Schlesinger has always grown on 7H11 plates and that's what he put into human macrophages for his virulence studies. As I have reviewed, many of the virulence studies done in the early days of TB biology, the strains were always grown on solid media.

Reply: We have now added details of strain maintenance and passage as pellicles on the surface of liquid media (line 468). We have noted the observation that strains grown on solid media have been used in virulence studies and passage of strains in liquid media can lead to loss of virulence.

Q3. You did not mention if the Erdman strain has esxX or esxY, which is missing from the original H37Rv genome sequence. I think you should reference the recent paper from Koleske et al 2025 (Science Advances paper). This region, when deleted, makes the strain hypervirulent compared to Erdman. Due to the PPE71 duplication it was missing when Stewart Cole's teams sequenced the ordered library of cosmids. Your technique of long reads is exciting and will become the standard of what others will follow.

Reply: We have now noted that esxX and esxY are present in the ErdmanTI genome, but it was not a major talking point for us because it was not a region of difference with respect to the previous reference strain of Erdman, nor with respect to H37Rv's current public annotation. There is one difference between Erdman and H37Rv though that is a presence if an insertion sequence upstream of PPE38 which might affect the expression of PPE38. We have cited the Koleske et al 2025 paper at line 235.

Q4. You should mention in the Kubica paper as well as the recent Mulholland paper that passage of strains in liquid media can lead to loss of virulence. In H37Rv strains, this is due to a loss of PDIM that can occur by mutations in many different genes. The Mulholland paper should be referenced and discussed with respect to your work. It is interesting that with the Erdman strain, you do not see any mutations that result in PDIM loss. This is interesting but not understood.

Reply: We acknowledge your comment. The Erdman strains were not routinely passaged on solid media, they were grown as a pellicle on the surface of liquid media. That was how strains were maintained before -70 freezers, passaged every month or two. That's how the stocks in the collection were prepared, grown as a pellicle, homogenized, then sonicated before being dispensed as detergent dispersed culture in liquid media is associated with mutation leading to loss of virulence. We have now mentioned this in the methods and referenced Kim TH, Kubica (1972) and Mulholland C V et al. (2024) papers to support this.

Q5. Please reference Dr. Erdman's original work of the isolation of this Mtb strain in your paper.

Reply: We found no primary publication by Dr. Erdman describing the strain's isolation. Instead, archival records attribute Mycobacterium tuberculosis Erdman (TMC 107) to William H. Feldman at the Mayo Clinic—isolated from human sputum in 1945 and deposited with the Trudeau Mycobacterial Culture Collection in 1946. Accordingly, we cite Miyoshi-Akiyama et al. (2012), which reports the complete annotated genome sequence of the Erdman strain.

REVIEWER #2 (COMMENTS FOR THE AUTHOR):

The manuscript by Modlin et al outlines a new reference genome for the Erdman reference genome for Mtb and some nice proposals of evolutionary trajectories that are observed from the related analysis.

Overall, the paper is robust with a lot of information. I don't have many major concerns.

Q1. My main issue is the 'story' that runs through. For example, the title is focused on the tandem repeat aspect, which is actually not a focus of the majority of the paper. Indeed, in the abstract and in many places in the results, it is difficult to find out which 'story' is the lead here: is it the reference genome issue that are corrected or the proposal of evolutionary paths that come from that? If the former, this is well supported and outlined in the paper. If the latter, much of this is proposed phenotypic outputs but without accompanying experimental data. I still think it is well supported from the genomic analysis, but it is not confirmed, which would require much more work.

Reply: Thank you for your feedback. Our initial approach was to update the reference genome of Erdman strain, but while updating the reference genome, we also uncovered variable numbers of tandem repeats (VNTRs) within subpopulations of the reference strain, suggesting

that the genome is not fixed but subject to phase-variable changes. They were not only observed in the laboratory reference strain, but also across a diverse set of clinical isolates, highlighting their potential role as dynamic elements of the *M. tuberculosis* genome. These findings provided a natural transition from updating the reference sequence toward examining how these tandem repeats may contribute to phenotypic diversity, including traits linked to virulence, persistence, and host adaptation. Hence, there is not just one story in our manuscript. We have tried to separate the two themes of our results into reference update added a transition between the two.

Q2. In line with this, the paper is very dense and difficult to follow. This is my area of specialty so I understood what was being explained but for someone with less expertise, this paper would be difficult to understand in many places. The combination of results and discussion make it hard to see what a direct result of this work is and what is a proposed explanation for some of the genomic features observed (but not validated).

Reply: We appreciate this very important comment. To improve clarity, we have now separated the findings and discussion by separating them into different paragraphs without disrupting the relationship between results and discussions.

Q3. Generally, the paper is a strong one with a lot of relevant information. I would suggest for readability it needs to be looked at to separate out the primary findings (and ensure they are clear in the title and abstract, which they currently are not) from the proposals for future investigation. This should include some simplification of text in the results/discussion.

Reply: Thank you for your feedback. We have now reflected both foci equally on the new abstract for the manuscript.

Q4. Minor points:

Line 460: A very short explanation of why the plasmid insertion is used here would be appreciated

Line 490: I think this is an incomplete sentence. This is an example of a few places in the text where there are grammatical/sentence structure issues that need another pass.

Reply: Thank you for the feedback. All these minor points have been addressed accordingly.

Re: mSystems01026-25R1 (**Updated Erdman reveals tandem repeat copy number is phase-variable and impacts *M. tuberculosis* adaptation across evolutionary timescales**)

Dear Dr. Faramarz Valafar:

Your manuscript has been accepted, and I am forwarding it to the ASM production staff for publication. Your paper will first be checked to make sure all elements meet the technical requirements. ASM staff will contact you if anything needs to be revised before copyediting and production can begin. Otherwise, you will be notified when your proofs are ready to be viewed.

Sincerely,
Zoe Dyson
Editor
mSystems